# Observers' motivated sensitivity to stigmatized actors' intent

**William A. Staples** **\*, Jason E. Plaks**

Psychology, University of Toronto, Toronto, Ontario, Canada

\* william.staples@mail.utoronto.ca

## Abstract

Does a harmful act appear more intentional–and worthy of opprobrium–if it was committed by a member of a stigmatized group? In two studies (N = 1,451), participants read scenarios in which an actor caused a homicide. We orthogonally manipulated the relative presence or absence of distal intent (a focus on the end) and proximal intent (a focus on the means) in the actor's mind. We also varied the actor's racial (Study 1) or political (Study 2) group. In both studies, participants judged the stigmatized actor more harshly than the non-stigmatized actor when the actor's level of intent was ambiguous (i.e., one form of intent was high and the other form of intent was low). These data suggest that observers apply a sliding threshold when judging an actor's intent and moral responsibility; whereas less-stigmatized actors elicit condemnation only when they cause the outcome with both types of intent in mind, more-stigmatized actors elicit condemnation when only one type, or even neither type (Study 2) of intent is in their mind. We discuss how these results enrich the literature on lay theories of intentionality.

## Introduction

Intentionality plays a crucial role in lay theories of morality [1–5]. For example, a central principle of Schein and Gray's [4] Dyadic Model of morality is that people condemn an actor to the extent that they perceive the actor as an intentional agent who harmed a vulnerable patient. Similarly, Malle et al.'s [3] influential theory of blame gives a prominent role to observers' assessment of whether the focal act was intentional or unintentional.

But what, in laypeople's minds, makes an act 'intentional'? For example, is it sufficient for the actor to hold a malevolent aim in mind at the moment of the act but then cause the outcome in an unplanned, serendipitous manner? Consider, for example, a man who aims a gun at his neighbor intending to kill him, but just before he pulls the trigger, the neighbor is so overcome with fear that he has a heart attack and dies. Alternatively, is it sufficient to perform the physical act with awareness and control, but in the absence of the malevolent aim? Consider, for example, a man who aims a gun at a target, practicing to kill his neighbor. He pulls the trigger with full concentration and awareness, but, just then, the neighbor unwittingly steps in front of the target and is killed by the bullet. How much moral responsibility would you assign to the man in each case?

**Data Availability Statement:** All data for both studies is held held in a public repository (Open Science Framework) at the following link: https://osf.io/8fqc7/.

**Funding:** J.E.P. received funding from the Natural Sciences and Engineering Research Council of Canada (fund # 508977).

**Competing interests:** The authors have declared that no competing interests exist.

Examples like these reveal that there are elements of lay models of intentionality that go beyond a simple, binary representation of 'intentional' versus 'unintentional'. This complexity is evident in the numerous, unstandardized ways in which researchers in the literature have operationalized intentionality. Some researchers have focused on whether the actor possessed conscious awareness of the (typically malevolent) desired end while performing the focal act. For example, Young and Saxe [5] presented participants with scenarios in which an actor knowingly put a toxic powder labeled either 'sugar' or 'toxic' into a co-worker's coffee. These researchers labeled the toxic condition 'intentional' and the sugar condition 'accidental'. Thus, the means (pouring the powder) was held constant, but the broader aim (sweeten coffee vs. murder coworker) was manipulated.

In contrast, other researchers have held the broader aim constant, while manipulating the actor's control over the means [e.g., 6,7]. For example, Pizarro, Uhlmann, and Bloom [7] presented participants with a scenario in which a woman wants to kill her husband by poisoning his food at a restaurant. In one condition, she pours poison into the food and he takes a bite, but the food merely tastes very bad and does not harm him. He then changes his order to a different dish which, unbeknownst to them both, contains a deadly allergen. He eats the new dish and dies. To summarize, some researchers have manipulated 'intent' by varying the actor's broader aim, whereas others have manipulated 'intent' by varying whether the death blow was performed with awareness and physical control.

## The PIDI framework

The Proximal Intent / Distal Intent (PIDI) framework aims to standardize the literature by specifying two distinct, at least partially independent dimensions in lay models of intent: *proximal intent* and *distal intent* [8]. Proximal intent refers to the actor's degree of conscious awareness and control while performing the focal action (e.g., pulling the trigger). That is, proximal intent refers to intent with respect to the *means*. Distal intent refers to the actor's conscious awareness of a broader aim beyond the physical motion itself (e.g., killing the neighbor). That is, distal intent refers to intent with respect to the *end*. It is important to note that we consider the presence or absence of proximal intent and distal intent as continua, not dichotomies. Thus, at a given moment, although an actor's distal intent may be more prominently in mind than her proximal intent, this does not imply that the proximal intent is entirely absent. Moreover, in certain cases, the key, focal distal intent (e.g., kill neighbor) may be low because another distal intent (hit target) is high. (Thus, the condition labeled "Distal Intent Low" is actually shorthand for the more cumbersome "*focal* Distal Intent Low".) Although we view proximal and distal intent as continua, in the present studies, we operationalize proximal and distal intent in a dichotomous fashion to permit the manipulation of both constructs in a standard, social psychological 2X2 full factorial design.

It is also important to note that distal intent differs from such concepts as 'desire', 'reason', 'motive', 'plan', and 'foreseeability' in that such concepts refer to pre-existing mental states that do not require an action. For example, one may hold a sexual *desire* toward another person, but never act upon it. Similarly, one may have a *reason* to act, a *motive* to act, and a *plan* to act, but never ultimately execute the act. Distal intent, in contrast, requires an act. Specifically, it refers to the actor's awareness, while the act is occurring, of the link between the act and the desired end. (For a fuller discussion of these distinctions, see 2.)

Several philosophers have described related distinctions. For example, Searle [9] distinguished between "prior intention" versus "intention-in-action", Duff [2] distinguished between "intended" versus "intentional" agency, and Brand [10] distinguished between "prospective" versus "immediate" intent. The psychological literature on such distinctions,

however, is relatively sparse. In early work, Ossorio and Davis [11] nominated *desire*, *knowledge*, *skill*, and *conventionality* as key ingredients in laypeople's understanding of intent. Later, Shaver [12], proposed *desire*, *beliefs about consequences*, and *beliefs about one's ability*. More recently, Malle and Knobe [13] provided more rigorous evidence for a five-part hierarchical model in which the agent's *belief* and *desire* contribute to the forming of an *intention*, which, in turn, is combined with *skill* and *awareness*. They provide evidence that the more of these elements are present in the actor's mind, the more participants rate the act as 'intentional'.

We view the PIDI framework as extending several of Malle and Knobe's elements by introducing proximal versus distal variants. For example, Malle and Knobe operationalized *intention* as "intending, meaning, deciding, choosing, or planning to perform the act" (p. 107). The PIDI model asks, what is the actor's subjective representation of the 'act'? For example, does he think he is 'pulling a trigger' or 'murdering his uncle'? Similarly, consider the concept of "desire." Malle and Knobe (1997) defined desire as "a wish for a particular outcome" (p. 106). The PIDI model asks, what *exact* desire is forefront in the actor's mind, pulling the trigger or killing the uncle? By now, numerous studies have provided evidence that perceivers are quite sensitive to these distinctions [14–16].

In more recent studies that relate more directly to the PIDI model, Levine and Leslie [17] distinguished between intent with respect to the means versus the end (the "means principle") and reported that even young children can readily distinguish between the two. However, because their aim was to investigate whether young children are capable of using means-end reasoning in their moral judgments, their studies did not vary the race/political orientation of the actor, nor did they use the proximal-distal, full factorial design. In a related vein, Laurent, Clark, and Schweitzer [18] reported that people tend to associate the noun 'intention' with an actor's broader goal (the end), but the adjective 'intentional' and the adverb 'intentionally' with the physical action itself (the means). However, because Laurent et al.'s aim was to shed light on a specific phenomenon (the 'side-effect effect'; 19), their experimental designs also did not involve the independent manipulation of the presence/absence of the two varieties of intent (See also 20).

## The basic PIDI effect

In several studies, however, researchers *have* independently manipulated both forms of intent in a 2 (proximal intent: high vs. low) X 2 (distal intent: high vs. low) design. Plaks, McNichols, and Fortune [14] asked participants to read about an actor ('Barbara') who desired to murder her husband, believed that she could murder him, formed a plan to murder him, and had the skill to murder him. In all cases, Barbara poured poison into her husband's dish and in all cases, the husband died. What differed between conditions, however, was the degree to which proximal and distal intent were present in Barbara's mind during the act. In one condition, both proximal and distal intent were high as the murder occurred according to plan: She skillfully poured the poison with awareness and control (proximal intent high), with the aim of murdering him (distal intent high). In another condition, proximal intent was high and distal intent was low, as she skillfully poured the poison, but in the service of a different goal that was not murdering her husband. In a third condition, distal intent was high and proximal intent was low (i.e., the case above when the poison caused the husband to change his dish, which ultimately caused his death). In the fourth condition, both forms of intent were low.

Participants rated the actor most responsible when both forms of intent were high in Barbara's mind, least responsible when both were low. In the two partially-intentional conditions (Distal Intent High / Proximal Intent Low and Distal Intent Low / Proximal Intent High), they rated the actor's culpability roughly mid-way between the two extremes. This pattern of two

main effects indicates that people generally consider proximal and distal intent to be at least somewhat independent dimensions that contribute additively to making an act appear more intentional. Further studies have demonstrated that this overall pattern extends to morally neutral acts and to prosocial acts [15].

**Moderators of the basic effect.** Beyond the main effects, researchers have identified psychological variables that interact with proximal intent, distal intent, or both. One example is psychological distance [21]. In one study, when the death blow was described as occurring 75 years ago (versus two weeks ago), participants placed more emphasis on distal intent (i.e. whether the malevolent goal was in the actor's mind at the moment of the act) than on proximal intent (i.e. whether the physical act was performed with awareness and control) [14]. An analogous pattern emerged when psychological distance was manipulated not via time, but via space (i.e., the event occurred in the participants' home country of Canada or in Russia) [15]. Other variables that have been found to interact with proximal and distal intent include individualist/collectivist culture and beliefs about free will/determinism [16,18]. In other words, certain beliefs and mindsets appear to shift observers' emphasis toward whether the actor's focus was on accomplishing a broader goal (distal intent) or on executing the physical motion (proximal intent). These shifts in emphasis lead to corresponding shifts in moral judgment of the action and the actor.

**The potential role of motivation.** The moderators of the PIDI effect examined thus far have generally involved 'colder' constructs such as mental representations of psychological distance. The present studies examined whether 'warmer,' motivated processes might similarly shift participants' relative emphasis toward proximal intent or distal intent. This approach is inspired by evidence that individuals' goals, desires, and preferences exert systematic, measurable effects on default reasoning processes [e.g., 22–25]. In other words, in certain circumstances, might moral judges be motivated to place greater weight on the actor's broader aim than on the physical means? In other cases, might moral judges be motivated to prioritize the physical means over the broader aim?

In the present studies, we leveraged potential prejudicial motivations that participants may hold by manipulating the stigmatized versus non-stigmatized status of the actor. We assumed that when an actor committed an act with only one form of intent in mind, participants would judge the actor more harshly if the actor was a member of a stigmatized racial (Study 1) or political (Study 2) group.

## Study 1

A voluminous psychological literature has demonstrated that people display a range of motivated and non-motivated biases when processing information about racial outgroups. Most of this research has focused on participants' perceptions of young, Black males and has detailed a range of biases uniquely associated with young, Black males [e.g., 26–28]. To the extent that such biases would be present in the Study 1 sample, how exactly would they be expressed? Three hypotheses appeared plausible to us, *a priori*. On the one hand, non-Black participants may subscribe to the stereotype that, compared to White individuals. Black individuals are more strongly governed by immediate, impulsive concerns, rather than abstract, rational aims [29–31]. In other words, this stereotype depicts Black actors as being primarily guided by proximal intent (i.e., execution of near-term, physical action), with less contribution from distal intent (i.e., a longer-term goal). According to this hypothesis, to the extent that observers expect Black (vs. White) actors to be heavily influenced by proximal intent, observers' calculations of moral responsibility will be more sensitive to the presence or absence of a Black actor's proximal intent than distal intent.

A second hypothesis is informed by the literature on psychological distance [e.g., 32]. Several studies have indicated that observers construe less familiar, psychologically distant target persons at a more abstract, 'big picture' level [33], resulting in more extreme dispositionism [34]. Thus, to the extent that observers construe stigmatized (vs. non-stigmatized) group members as more psychologically distant, they may place greater weight on the presence or absence of distal intent than on the presence or absence of proximal intent. Such a pattern would accord with Plaks, McNichols and Fortune [14; Study 2], who found such an effect when psychological distance was manipulated via time.

A third hypothesis is that Black people will be subjected to a generalized derogation effect [35] that does not track subtle distinctions in the actor's proximal or distal intent. Should this be the case, non-Black observers will more readily condemn a Black actor (and give the White actor the comparative benefit of the doubt) across the board, in all four conditions. Study 1 investigated which of these three hypotheses would fit best with the data.

## Method

**Ethics statement.**   Both studies received approval from the University of Toronto's Social Sciences and Humanities Research Ethics Board. Prior to beginning, all participants were provided with information about the study and were given the option to consent or decline to participate. A full briefing was provided at the end of the study in which participants could direct follow-up questions to the principal investigator. Following the debriefing, participants were provided with the opportunity to refuse to allow their data to be used. Study 1 was approved under the designation REB Protocol # 44591.

**Participants.**   Three hundred and ninety-seven participants (192 males and 203 females, 2 undisclosed, median age bracket of 30–39) were recruited via Amazon's Mechanical Turk between October 24th and November 1st, 2014. Ethnic breakdown: 57% White, 21% South Asian, remainder four other ethnicities. This sample size was determined by comparison with similar studies that investigated a Proximal Intent X Distal Intent X moderating variable design [14–16], and have yielded greater than 80% observed power.

*Scenarios*. We used scenarios presented by Plaks and Robinson [15]. In the scenarios, Alex wants to kill his ex-girlfriend, Linda, by tying her up with a rope and drowning her in a lake. What varies among the four scenarios is the degree to which distal intent and/or proximal intent are present in Alex's mind. In the Both Present scenario, Alex thinks about killing Linda while he carries out his plan to deliberately tie her up and throw her into the lake. In the Distal Intent High / Proximal Intent Low scenario, Alex thinks about killing Linda while Linda is killed, but Linda's death occurs in an unintended manner (Linda trips, hits her head on the dock, and falls into the lake while attempting to flee). In the Distal Intent Low / Proximal Intent High scenario, Alex's mind is not focused on the goal of killing Linda; rather, he thinks about the best method to untangle a rope when he deliberately pulls the rope, causing Linda to fall into the lake and drown. In the Both Absent scenario, Alex's mind is not focused on the goal of killing Linda. Rather, Alex thinks about the best method to untangle the rope at the moment when Linda dies because she trips, hits her head on the dock, falls into the lake, and drowns.

*Actor race manipulation*. Roughly half of the participants (n = 198) were randomly assigned to view the actor (Alex) paired with a photo of a young Black man; the remainder (n = 199) saw a photo of a young White man.

*Primary dependent variable*: *Moral Judgment Index*. The moral judgment items (reported in previous studies, [14–16] were: "To what extent were Alex's actions intentional?", "How much moral responsibility does Alex deserve for what happened?", "Either negatively or positively,

how strongly should Alex be judged?", "How much blame or praise should Alex receive as a result of their actions?", "Do you think that Alex will receive some sort of punishment or reward for their actions?". Participants provided ratings on 0–5 scales. Because the items were highly intercorrelated ($\alpha$ = .88), we aggregated them into a single moral judgment index.

*Measures of prejudice*. To assess potential moderating variables, we included three measures of individual differences related to prejudice. The *Attitudes Toward Blacks Scale (ATBS)* developed by Brigham [36] examines respondents' level of explicit prejudice toward Black Americans. The *External Motivation to Respond Without Prejudice Scale (EMS)* [37] assesses the degree to which respondents want to appear unprejudiced to observers. The *Internal Motivation to Respond Without Prejudice Scale (IMS)* [37] assesses the degree to which respondents want to appear unprejudiced to themselves because it is consistent with personal values.

*Procedure*. Participants provided written informed consent as well as demographic information, including age, gender, and ethnicity. Then participants completed the ATBS, EMS, and IMS. Next, participants were told that they would move on to a second, unrelated study. At this point, participants were randomly presented with one of the four Alex/Linda scenarios on the computer screen. For half of the participants, Alex was paired with a photo of a young Black man and the other half of a young White man. Participants provided ratings on the moral judgment items. The study concluded with a written debriefing.

## Results

To assess the effect of actor race and the presence/absence of proximal and distal intent on participants' judgments of the actor's moral culpability, we performed a regression analysis with the moral judgment index ($\alpha$ = .88) as the dependent variable and Distal Intent, Proximal Intent, Actor Race, all two-way and three-way interactions entered as predictor variables. All variables were mean-centered.

First, there was not a significant main effect of Actor Race ($\beta$ = -.084, $t(7, 389)$ = -.92, $p$ = .359) indicating that, collapsed across all intent conditions, participants did not display an overall tendency to judge the Black actor more harshly than the White actor. This analysis did reveal a main effect for Distal Intent ($\beta$ = .525, $t(7, 389)$ = 5.147, $p < .001$). In other words, participants rated the actor more culpable when Distal Intent was prominent in his mind than when it was not. The analysis further revealed a marginal actor race by Proximal Intent interaction ($\beta$ = .213, $t(7,389)$ = 1.9, $p$ = .058).

Although the Actor Race by Proximal Intent interaction was not robustly significant, we nonetheless performed simple effects analyses within each of the four intent conditions to test our more targeted hypotheses. When both Proximal and Distal intent were present in the actor's mind, there was no effect of Actor Race ($\beta$ = -.104, $t(1, 96)$ = -1.028, $p$ = .306). Likewise, when both Proximal and Distal Intent were low in the actor's mind, there is also no effect of Actor Race ($\beta$ = -.081, $t(1, 98)$ = -0.807, $p$ = .422). In other words, when the act was blatantly intentional or blatantly unintentional, participants did not distinguish between whether the actor was Black or White.

However, when Distal Intent was high and Proximal Intent was low, participants judged the Black actor to be more morally culpable than the White actor ($\beta$ = -.248, $t(1, 98)$ = -2.536, $p$ = .013). When Proximal Intent was high and Distal Intent was low, the effect was not significant, ($\beta$ = .174, $t(1, 97)$ = 1.744, $p$ = .084) (See Fig 1).

In summary, this pattern provides preliminary evidence that when it comes to weighing an actor's level of intent and moral responsibility, Black actors are subject to a different moral calculation than are White actors. When Alex had the aim of killing Linda in mind (distal intent high) at the moment when Linda died via an unexpected, alternative means (proximal intent

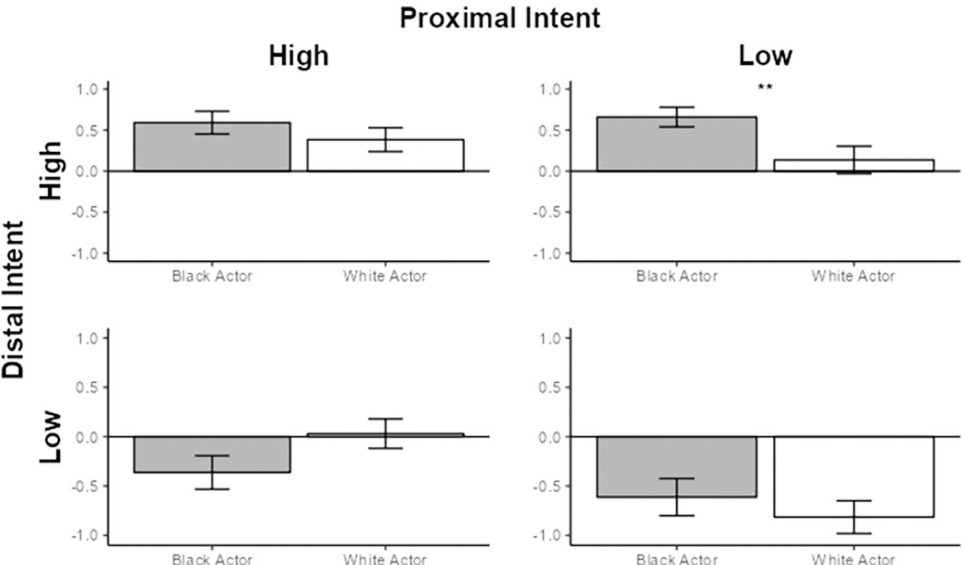

**Fig 1. Mean moral judgment ratings of the White and Black actor in all four intent conditions.** Y-axis indicates mean-centered Moral Judgment Index score. Asterix denote significance.

low), participants (93.5% of whom were non-Black) rated Alex more negatively if he was Black than if he was White.

**Effects of IMS, EMS, and ATBS.**   The IMS ($\alpha = .79$), EMS ($\alpha = .90$), and ATBS ($\alpha = .91$) scales all displayed good reliability. However, analyses reported in the Supplementary Materials (S1 File) revealed that none of these measures moderated the effects reported above in a manner relevant to the present hypotheses.

## Discussion

The Study 1 data provide initial evidence that observers' hold stigmatized (compared to non-stigmatized) actors to a different standard of intentionality. Non-Black participants judged the Black actor more harshly than the equivalent White actor only in the condition when the actor's distal intent was high (the broader intent to kill was in his mind), but proximal intent was low (the outcome was achieved in a serendipitous manner).

This pattern fits best with our second hypothesis. The first hypothesis predicted anti-Black bias to be especially evident in the Distal Intent low / Proximal Intent high condition, on the basis of stereotypes regarding higher impulsivity [30,31]. This hypothesis was not supported. The third hypothesis predicted anti-Black bias to be generally evident in all conditions. This was also not supported. Instead, the observed pattern is consistent with the ideas that (a) stigmatized group members are viewed as more psychologically distant than are non-stigmatized members [e.g., 34] and (b) high (versus low) psychological distance encourages an emphasis on distal intent [14].

The fact that differential judgments of the Black versus White actor only emerged when the actor's degree of intentionality was ambiguous, suggests that observers' motivated biases operate with 'reality constraints' [38], whereby the presence of ambiguity fosters plausible deniability of biased processing [39]. Indeed, it is a central principle of motivated cognition that people generally engage in such biased processing only when there is little danger of being exposed as biased or delusional. For example, Kunda [38], in the foundational text of the motivated cognition literature, noted, "Clearly, then, if directional goals do exert an influence on reasoning,

this influence is limited by people's perceptions of reality and plausibility" (p. 491). This concept has been found to characterize motivated cognition in countless subsequent studies, including those on 'social judgeability' [40,41] and system justification [42,43].

One potential limitation to this study is that the data were collected in 2014. It is an interesting open question whether the effect reported here would replicate today. On the one hand, more widespread recognition of racial inequalities introduced by the Black Lives Matter movement may lead contemporary American participants to display less bias toward Black actors. On the other hand, there is evidence of at least some backlash against such movements in the United States, as well as data indicating that racial attitudes remain largely unchanged from 10 years ago [44]. Thus, it remains to be seen whether the effects reported here would differ with a contemporary sample.

An additional important limitation of this study is the lack of Black participants in the sample. A more robust presence of Black participants would allow us to test whether the effects observed in Study 1 extend to racial outgroups in general or are limited to Black target persons. (i.e. Do Black observers similarly judge a White actor more harshly only in the Distal Intent Higher condition?) An additional limitation is that the sample size did not permit an examination of how participants' intersectional identities may have altered the results. For example, might the effects we observed differ depending on whether participants were male or female? Whether the actor was male or female? We encourage future researchers to investigate such questions. We turn next, however, to a different class of stigmatized group: political anti-partisans. If motivated weighting of proximal and distal intent is truly a pervasive phenomenon, it should occur in other types of intergroup settings.

## Study 2

In recent years, political polarization has increasingly legitimized various forms of political intolerance in the United States, by members of both the left and the right [e.g., 45–48]. As such, when a political anti-partisan causes a harmful outcome with an ambiguous degree of intentionality, people may be especially motivated to judge the actor harshly, while giving the benefit of the doubt to an actor who shares one's view.

Study 2 followed a design that was akin to that of Study 1 but held the actor's race constant and substituted the actor's political self-identity (conservative or liberal). Based on Study 1's finding that the actor's racial identification only affected participants' judgment in the condition when distal intent was high and proximal intent was low, we hypothesized that in this condition, relatively liberal participants would judge a conservative actor more harshly than a liberal actor and relatively conservative participants would judge a liberal actor more harshly than a conservative actor.

### Participants

One thousand and fifty-four (468 males and 468 females, 33 nonbinary, 8 undisclosed, mean age = 35.1, mean political orientation = 2.28, indicating slightly liberal) were recruited via Prolific.com between August 30th and September 7th, 2022. Ethnic breakdown: 720 White, 107 South Asian, 98 Hispanic, 76 Black, 44 biracial, 9 other races. In Study 1, the vast majority of participants (93%) were non-Black and, thus, would be more likely to view Black target persons as members of a comparatively stigmatized group. In Study 2, we assumed that although the political orientation distribution would be somewhat skewed toward the liberal side, there would be a substantial subset of conservative participants. This meant that, to investigate whether the effect observed in Study 1 extended to both liberal-leaning and conservative-leaning participants, we needed to roughly double the sample size.

**Actor political orientation manipulation.** Participants provided written informed consent as well as demographic information. Participants were shown a series of eight social media posts in randomized order. Four of these posts contained statements that were highly indicative of the actor's symbolic political orientation and four were neutral statements intended to make the manipulation less obvious (see Fig 2). Politically aligned posts were created by referring to polarizing political issues relevant to the North American political sphere at that time. All stimuli can be found in S2 Appendix.

**Scenarios.** We used scenarios presented by Plaks and colleagues [14]. These scenarios are highly similar to those used in Study 1, following the same Proximal Intent / Distal Intent structure, but depicting a man killing his uncle to gain a sizable inheritance (see S1 Appendix in S1 File).

*Participants' Political Orientation.* To measure participants' political orientation (for the purpose of determining whether a given political tweet represented an attitude-consistent versus -inconsistent view), we used the Attitude-Based Political Conservatism (ABPC) questionnaire (for validation information, see 41).

We divided our sample into tertiles with the bottom tertile (the lowest third of ABPC scorers) designated "liberals" (n = 465) and the top tertile (the highest third of ABPC scorers) designated "conservatives" (n = 464). By removing moderates, we aimed to isolate the more politically extreme participants, i.e., those for whom it is easier to identify positions expressed by the actor that would be clearly attitude-consistent and -inconsistent.

## Moral judgment index

To improve the psychometric rigor of our dependent measure, we created and validated the 6-item Moral Judgment Index (MJI). A factor analysis of the MJI using the Study 2 data revealed two factors, which we labeled 'intent' and 'valence'. In other words, the intent factor contained items such as "To what extent were JG's actions *intentional*?". The valence factor focused more on generalized goodness/badness judgments as in "How *positively* should JG be judged?" The full two-factor MJI is included in S3 Appendix.

Analyses of this measure indicated that item 4 ("How *positively* should 'actor' be judged?") and 6 ("Even if no one ever finds out what they did, 'actor' will *get what they deserve*.") did not capture meaningful variance in respondent scores (see S4 Appendix). Additionally, these items did not correlate with other items in the measure. To minimize predictive error, we chose to remove these items. Thus, the resulting dependent measure was composed of four items ("To what extent were the actor's actions intentional?"; "To what extent can one say that the actor did what they did on purpose?"; "How negatively should the actor be judged?"; "How much blame should actor receive as a result of their actions?")

Further analyses revealed that the MJI data were negatively skewed (skewness values provided in S5 Appendix). To avoid violating statistical assumptions, we used a square transformation to reduce the absolute skewness of the data. (For more on square transformation, see 49) All variables were mean centered.

## Results

**Political orientation.** Participant political orientation was measured using the ABPC questionnaire [50]. On the ABPC, participants rate their level of agreement/disagreement (on 1–5 scales with 1 indicating "Strongly disagree" and 5 indicating "Strongly agree") to 33 statements that capture defining features of contemporary left-to-right political identity (e.g., "A large government is necessary to ensure that all our country's citizens are taken care of."; "If a pregnant woman believes abortion is the best choice for her, she should not have to defend

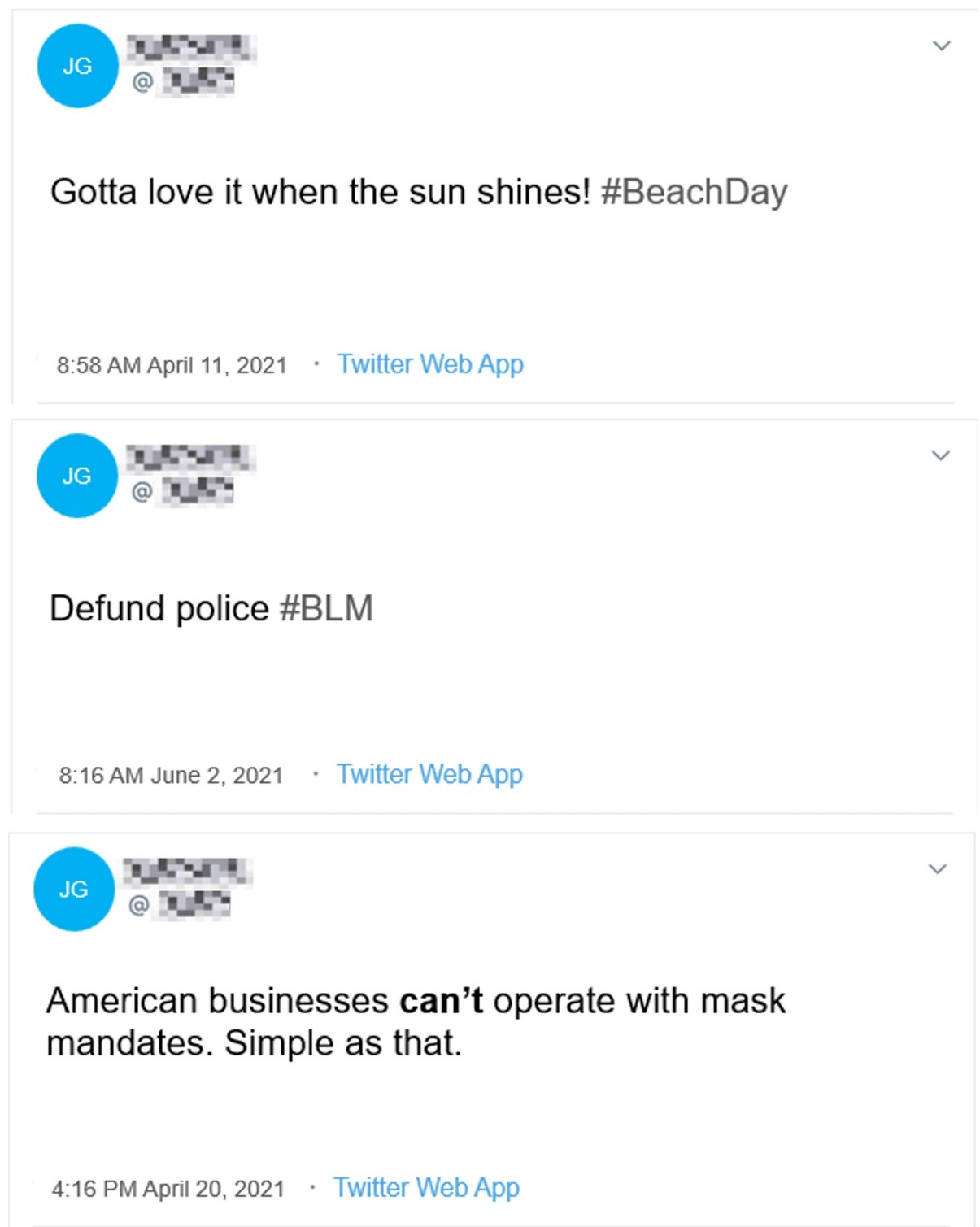

**Fig 2.** One of the four neutral posts (top), one of the four liberal posts (middle), and one of the four conservative posts (bottom).

that choice to anyone."). (For the full ABPC scale, see S6 Appendix. Note, the ABPC made available in S6 Appendix uses the original scale range of 1–7, our implementation uses a range of 1–5 to better accommodate participants using smartphones).

The scale displayed good reliability ($\alpha$ = .96, mean = 2.26, $\sigma$ = 0.9). As noted, we divided the sample into tertiles to remove politically moderate participants from the analyses. We did this because our design required clear identification of stigmatized vs. non-stigmatized (anti-partisan vs. pro-partisan) actors. For moderates, however, it is ambiguous how a liberal actor and/or a conservative actor would be stigmatized. The range of scores in the relatively liberal tertile was 1.12–1.79 (mean = 1.52, $\sigma$ = 0.17). The range of scores in the relatively conservative tertile was 2.55–4.55 (mean = 3.19, $\sigma$ = 0.46). These data indicate that the sample skewed toward the liberal end of the scale. As such, the more conservative participants tended to mark values near the mid-point of the scale. For this reason, we operationalize participants' political orientation in a relative, rather than absolute fashion.

**Main analyses.**   To assess the effect of actor political affiliation, participant political affiliation, and the relative presence/absence of proximal and distal intent on participants' judgments of the actor's moral culpability, we performed a regression analysis with the moral judgment index ($\alpha$ = .83) as the dependent variable and Distal Intent, Proximal Intent, Actor Political Affiliation, Participant Political Affiliation, all two-way and three-way interactions entered as predictor variables.

This analysis revealed main effects for Proximal Intent ($\beta$ = .41, $t(11, 917)$ = 7.27, $p < .001$) and Distal Intent ($\beta$ = .164, t(11, 917) = 2.905, $p = .004$), such that (unsurprisingly) participants rated the actor more culpable when either Proximal or Distal intent was present in his mind compared to when absent. The analysis further revealed a main effect of Participant Political Orientation ($\beta$ = -0.328, $t(11, 917)$ = -4.68, $p > .001$), indicating that the relatively conservative participants generally judged the actor more harshly, regardless of which attitudes the actor expressed. Finally, we observed a main effect of Actor Political Orientation ($\beta$ = -0.169, $t(11, 917)$ = -2.522, $p = .012$.), indicating that participants generally judged the conservative actor more harshly than they judged the liberal actor.

The analysis also revealed the following two-way interactions: Distal Intent X Participant Political Orientation ($\beta$ = .151, $t(11, 917)$ = -2.74, $p = .03$), Proximal Intent X Participant Political Orientation ($\beta$ = .129, $t(11, 917)$ = 1.869, $p = .062$), and Participant Political Orientation X Actor Political Orientation ($\beta$ = .525, $t(11, 917)$ = 6.308, $p < .001$). This latter interaction indicates that participants generally judged the actor whose views were inconsistent with their own views more harshly than the actor who shared their own views.

Most germane to our hypotheses, the analysis also revealed theoretically meaningful three-way interactions for Participant Political Orientation X Actor Political Orientation X Proximal Intent ($\beta$ = -0.166, $t(11, 917)$ = -2.303, $p = .022$) and Participant Political Orientation X Actor Political Orientation X Distal Intent ($\beta$ = -0.209, $t(11, 917)$ = -2.886, $p = .004$).

To probe these interactions more closely, we performed regression analyses separately for the relatively liberal and relatively conservative sub-samples, followed by simple effects analyses within each of the four intent conditions.

**Main effects for relatively conservative participants.**   For participants with ABPC scores in the top (conservative) tertile, we observed a significant main effect of Proximal Intent ($\beta$ = 0.195, $t(7, 456)$ = 2.574, $p = .01$) and a marginal main effect of Actor Political Orientation ($\beta$ = -0.142, $t(7, 456)$ = -1.906, $p = 0.057$). This indicates that, overall, relatively conservative participants, if anything, displayed a tendency to judge the liberal actor more harshly than the conservative actor, although this effect must be interpreted with caution.

**Simple effects for relatively conservative participants.**   Simple effects analyses within each of the four intent conditions (Both Present, Distal Intent High / Proximal Intent Low,

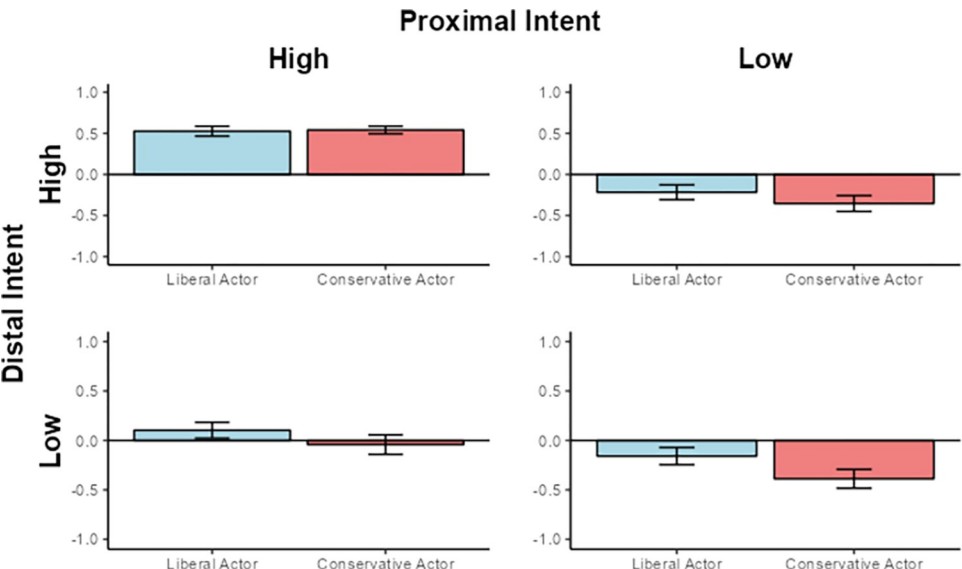

**Fig 3. Judgments of the liberal and conservative actor by relatively conservative participants in all four intent conditions.** Y-axis indicates mean-centered Moral Judgment Index score.

Proximal Intent High / Distal Intent Low, Both Absent) for relatively conservative participants revealed only marginal effects in the Both Absent ($\beta$ = -0.167, $t(1, 120)$ = -1.856, $p$ = .066) and Distal Intent High / Proximal Intent Low conditions ($\beta$ = -0.172, $t(1, 104)$ = -1.784, $p$ = .077). The direction of the effects indicated that in these two conditions, relatively conservative participants, if anything, judged liberal actors more harshly than conservative actors (see Fig 3).

**Relatively liberal participants.** Analogous analyses within the relatively liberal sub-sample revealed a main effect of Proximal Intent ($\beta$ = .37, $t(7, 457)$ = 4.723, $p < .001$) and a main effect of Actor Political Orientation ($\beta$ = .442, $t(7, 457)$ = 5.675, $p < .001$). These main effects were qualified by a Distal Intent X Actor Political Orientation interaction ($\beta$ = -0.244, $t(7, 457)$ = -2.659, $p$ = .008), indicating that relatively liberal participants judged the conservative actor more harshly than the liberal actor when Distal Intent was absent compared to when Distal Intent was present.

**Simple effects for relatively liberal participants.** To understand the pattern in greater detail, we examined relatively liberal participants' judgments of the liberal versus conservative actor within each of the four intent conditions. In the Proximal Intent High / Distal Intent Low condition, we observed a significant effect for Actor Political Orientation ($\beta$ = .225, $t(1, 128)$ = 2.62 $p$ = .01). In other words, when the actor produced the outcome without the malevolent aim in mind, but with a controlled muscle movement, relatively liberal participants judged the conservative actor more harshly than the liberal actor.

In addition, in the Both Absent condition a similar significant effect emerged ($\beta$ = .368, $t(1, 128)$ = 4.50, $p < .001$). In other words, even when the actor caused the outcome with neither the malevolent aim in mind nor a controlled muscle movement, relatively liberal participants judged the actor more harshly if he was conservative than if he was liberal (see Fig 4).

It should be noted that the effect observed in the Both Absent condition exceeded our expectations. In Study 1, the effect of the actor's race only emerged in conditions when the actor's intent was ambiguous (i.e., one type of intent was high and the other was low). The Both Absent conditions, in contrast, was created to be unambiguous; the act was described as clearly accidental. Nonetheless, relatively liberal participants still rated the conservative actor more harshly than the liberal actor in this condition. This may have occurred for two reasons.

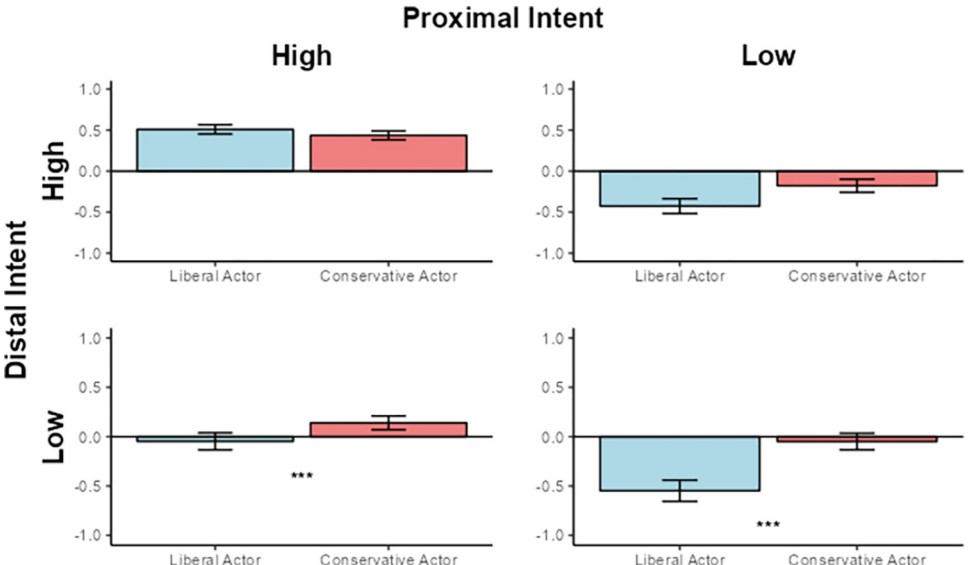

**Fig 4. Judgments of the liberal and conservative actor by relatively liberal participants in all four intent conditions.** Y-axis indicates mean-centered Moral Judgment Index score. Asterix denote significance.

First, the Both Absent condition scenario may not have been interpreted by participants to be as unambiguous as we, the experimenters, thought it to be. In other words, participants may have, despite our best efforts, found opportunities in the text to link the actor to malevolent intent and moral responsibility. Second, political intergroup bias may be considered more morally justifiable than racial intergroup bias. Recent data suggest that Americans increasingly view prejudice toward one's political outgroup as the number one form of *legitimate* prejudice [e.g., 45]. As such, participants may have felt less need to express their antipathy behind the veneer of objectivity.

Of note, evidence of such derogation was clearer in the relatively liberal sub-sample than in the relatively conservative sub-sample. This may be due to the liberal skew in the overall sample. To generate the two groups, we divided the sample into tertiles (instead of the midpoint of the ABPC's 1–5 scale) to achieve the roughly equal-sized groups needed to satisfy statistical assumptions. However, as depicted in Fig 5, the fact that the sample was positively skewed meant that the liberal tertile was composed mainly of those scoring between 1 and 2 and the conservative tertile between 2.5 to 4, with few scoring higher than 4. Thus, the two groups may more accurately be described as 'liberals' versus 'centrists'. This relative lack of 'true' conservatives in the sample provides an explanation for why the liberal group displayed stronger evidence of motivated cognition. This raises the possibility that the significant effects reported here, in fact, *understate* the magnitude of the differences. A future sample that included 'true' conservatives (i.e. those who score near the conservative end of the scale) may reveal a pattern that represents an even more stark mirror image of the pattern displayed by relative liberals in the present study.

Another limitation of the study is its exclusive North American context. Political orientation is operationalized and expressed in different ways throughout the world, with noteworthy differences from the North American context. For example, other nations and contexts may be less politically polarized; as such, the degree of 'enemyship' driving the Study 2 effects may be lower in other nations. We encourage future researchers to replicate the Study 2 design, but adapted to different national contexts.

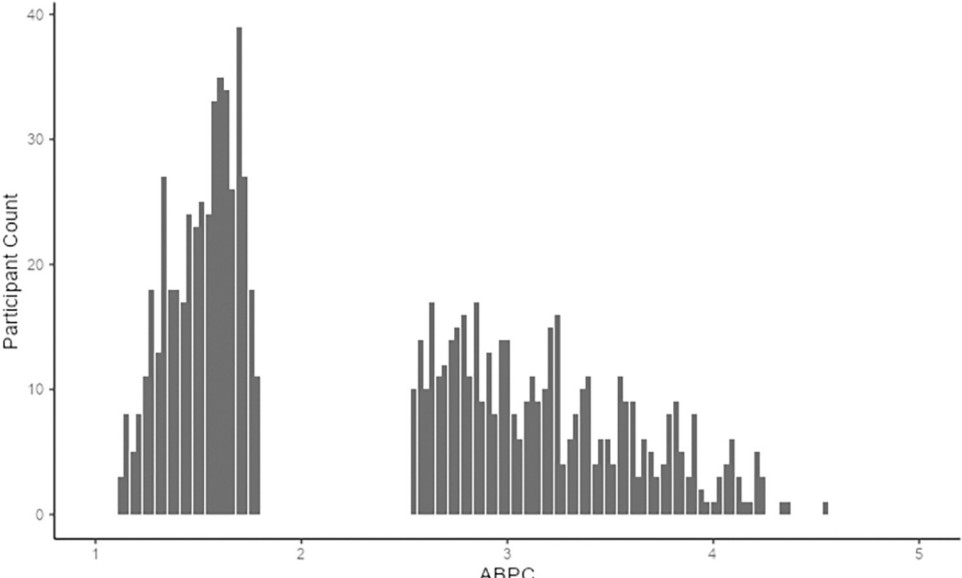

**Fig 5.** Distribution of participants in the liberal tertile (left side of the graph) and conservative tertile (right side of the graph).

## General discussion

In Study 1, non-Black participants judged a Black actor more harshly than a White actor only in the condition when the actor had the malevolent goal in mind but caused the homicide through coincidence. In other words, non-Black participants did not generally derogate the Black actor; instead, they appeared to judge the Black actor more harshly only when his intent was ambiguous. These results open the door to further investigation of how these results may differ as a function of participants' or actors' intersectional identities. In Study 2, political partisans generally rated a politically stigmatized actor more harshly than a non-stigmatized actor, but further analyses revealed that this was especially true of highly liberal participants and when the actor's distal intent was absent.

### Implications for understanding lay models of intent

A foundational principle of criminal law is "*actus reus non facit reum nisi mens sit rea*" ("The act is not guilty unless the mind is guilty."). But what constitutes a 'guilty mind'? Previous studies have examined variables that predict observers' degree of sensitivity to whether the actor committed the act 'intentionally' or 'unintentionally' in a binary sense [51–53]. In support of a binary model, we found that, in general, participants judged the actor with both forms of intent in mind to be more morally responsible than the actor with neither form of intent in mind. The results also indicate, however, that participants hold not only a unidimensional model of intent, but a model that is sensitive to two at least partially independent dimensions. Moreover, these data indicate that observers' motivation may subtly shift the amount of weight they place on one dimension versus the other.

Additionally, we note that in the present studies we operationalized proximal and distal intent in a dichotomous manner (high vs. low). We believe, however, that people often mentally represent the presence/absence of proximal and distal intent as continuous dimensions. Thus, the experimental conditions in the present studies refer to relative, not absolute differences. For this reason, we suggest that when one form of intent is more accessible in the actor's mind, the other form of intent is rarely entirely absent.

## Legal implications

These studies, along with [16,18,54], provide social psychological evidence for concepts that have long been discussed by philosophers, ethicists, and legal theorists [e.g., 2,55,56]. Many moral and legal judgments depend on inferences about the actor's mental state [55,57]. Thus, the legal definitions of numerous criminal offenses, including homicide, contain not only the act and its consequences (*actus reus*) but also the concomitant beliefs and goals of the actor (*mens rea*) [2,58]. Juries are composed of laypeople; thus, a fuller understanding of commonplace folk beliefs about *mens rea* may contribute to the creation of legal codes that are considered legitimate by the population at large. Moreover, although legal codes influenced by English Common Law tend to emphasize premeditation over behavioral control [e.g., 18 U.S.C. §1111, 1999, cited in 59], individual actions may vary in their degree of premeditation. For this reason, some legal scholars have suggested that English Common Law-influenced legal systems are not ideally equipped to handle cases such as those described here, in which intentionality is ambiguous because distal and proximal intent are decoupled [59,60]. The present studies document that observers (and potential jurors) are sensitive to whether the actor is a member of a stigmatized group. This perception may activate the motivation to 'throw the book' at the actor versus give the actor the benefit of the doubt, which, in turn, may lead observers to disambiguate ambiguity about intent in systematic and predictable ways.

## Ecological validity

We acknowledge that the stimulus scenarios we presented to participants might appear low in ecological validity. We suggest, however, that these types of scenarios do occur and present fascinating challenges to juries and legal thinkers. Consider, for example, the case of South African athlete, Oscar Pistorius. No one disputes that on the night of February 13, 2013, he shot and killed his girlfriend, Reeva. What was disputed was his precise intent at the moment he pulled the trigger. Pistorius claimed that although he pulled the trigger with awareness and control (proximal intent high), his aim was to kill someone on the other side of a door whom he believed to be an intruder. (It turned out to be Reeva.) Thus, to put the scenario into the language of the PIDI model, his distal intent was not to kill Reeva. At that moment, a different distal intent (kill the intruder) was at the forefront of his mind. This case split world opinion and the South African legal system, moving through several levels of appeals until it reached the nation's Supreme Court. Pistorius ultimately received a sentence (seven years) that was significantly lower than the maximum. Numerous legal theorists have raised these types of 'causally deviant' scenarios [55,58,60] precisely because they force scholars to define such legal terms as 'purposefully,' 'knowingly,' 'recklessly,' and 'negligently' more precisely, in psychological terms.

Moreover, given that proximal intent and distal intent are continua, *all actions* can be said to fall somewhere in the proximal intent–distal intent two-dimensional space. Thus, many legal and philosophical traditions [59,61] argue that contrived–though carefully crafted–scenarios can be valuable because they begin to isolate more precisely the specific building blocks of laypeople's moral judgment. By analogy, consider that researchers in visual perception have for decades presented stimuli (e.g., random-seeming patterns of lines and shapes; squares of color) that are dramatically impoverished compared to reality. Yet, they do so, because they aim to identify the building blocks of more complex forms of perception. So too, moral psychologists have created a host of scenarios that are purposely simpler than reality in order to identify analogous foundations of everyday moral cognition [e.g., 62].

## Cultural differences

A limitation of the present studies is that the samples were drawn from North American populations. Previous studies have indicated that North American participants tend to place more weight on the presence or absence of distal intent than proximal intent, whereas participants in South Asian and East Asia tend to place more weight on proximal intent than distal intent [16]. Building on work by Savani and colleagues [63], Plaks et al. [16] argued that Americans' "disjoint" model of agency leads them to view actions as emanating from the individual's idiosyncratic "preferences, beliefs, and goals" [64]. In contrast, non-Western societies' "conjoint" model of agency lead them to view actions as "responsive to the situation, to social roles, and to expectations of other individuals" [64]. In other words, North Americans are more likely to view an action as a choice. Construing an action as a choice suggests construing it as a means to accomplishing the individual's desired end. This perspective thus places more focus on the actor's distal intent. In contrast, South and East Asians are more likely than North Americans to view individual actions as responses to larger situational or normative forces. In such a model, the actor's idiosyncratic desires hold less relevance. With distal intent thus minimized, non-Western participants' judgments of responsibility should place greater weight on other components of intentionality—including proximal intent. Although Plaks et al. [16] measured participants' culture, they did not manipulate whether the actor was a member of a stigmatized group. We encourage future researchers to do so. Given the present finding that North Americans judged the stigmatized actor more harshly than the non-stigmatized actor when distal intent was most prominent in the actor's mind, one intriguing hypothesis is that non-Western participants will judge the stigmatized actor more harshly when *proximal* intent is most prominent in the actor's mind.

## Additional open questions

The present studies were not designed to assess additional psycho-legal concepts that likely contribute to assessments of an actor's intent, such as *foreseeability* [2]. Nor did these studies examine other mental processes that observers may extract from an actor's behavior. Indeed, several studies have demonstrated that observers also make spontaneous inferences about a range of mental states including beliefs, desires, goals, and traits [e.g., 65,66]. Future researchers would do well to systematically examine how inferences about proximal and distal intent relate to inferences of other psychological forces operating in the actor's mind.

The studies reported here involve an actor performing an act that is clearly harmful. Thus, a reasonable question is whether people apply the PIDI framework to understanding beneficial acts. Plaks and Robinson [15] and Plaks et al. [16] conducted studies in which the actor caused a positive outcome with both forms of intent high, distal intent higher, proximal intent higher, or both low. Although more studies are needed to build a larger body of evidence, the existing data indicate that people generally apply the PIDI framework to positive acts in much the same way as they do to negative acts. In none of those studies, however, did the researchers manipulate whether the actor was a member of a stigmatized group. Given the data reported here, one might hypothesize a mirror image pattern for positive acts: When an actor performs a prosocial act with distal intent high and proximal low, participants will praise her more if she is a member of a non-stigmatized group than if she is a member of a stigmatized group. We encourage future researchers to test this hypothesis.

Another avenue for future research involves transgressions that are less extreme than homicide, e.g., white collar crimes. Do people apply the PIDI framework in an equivalent way to, say, insider trading? This remains an unexamined question. An additional future direction involves action versus inaction. To our knowledge, none of the studies using the PIDI

framework have manipulated whether the act was an omission or a commission. Yet, this distinction has been demonstrated to be central to laypeople's moral calculus [e.g., 61,67,68]. Do people apply the PIDI framework differently for omissions versus commissions? This represents another promising line of future research.

## Supporting information

**S1 File. OMS Supporting information.**
(DOCX)

## Acknowledgments

The authors acknowledge the valuable insights provided by members of the University of Toronto Motivation and Social Cognition laboratory.

## Author Contributions

**Conceptualization:** William A. Staples, Jason E. Plaks.

**Data curation:** William A. Staples.

**Formal analysis:** William A. Staples.

**Funding acquisition:** Jason E. Plaks.

**Investigation:** William A. Staples, Jason E. Plaks.

**Methodology:** William A. Staples, Jason E. Plaks.

**Project administration:** Jason E. Plaks.

**Resources:** Jason E. Plaks.

**Supervision:** Jason E. Plaks.

**Validation:** William A. Staples.

**Visualization:** William A. Staples.

**Writing – original draft:** William A. Staples, Jason E. Plaks.

**Writing – review & editing:** William A. Staples, Jason E. Plaks.

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
