## [Decision Letter · Decision Letter 0]

19 Mar 2024

PONE-D-23-26287Observers’ Motivated Sensitivity to Outgroup Actors’ IntentPLOS ONE

Dear Dr. Staples,

Thank you for submitting your manuscript to PLOS ONE. After careful consideration, we feel that it has some merit but falls short of PLOS ONE’s publication criteria as it currently stands. Therefore, we invite you to submit a revised version of the manuscript that addresses the points raised during the review process.

Crucially, your revision should incorporate the following: 1. Collect more data or reframe the paper in line with Reviewer 1's recommendation.2. Correct the error pointed out by Reviewer 1 in their point 2 (having to do with the mismatch between the presented scenario and the appendix description).3. Take heed of Reviewer 1's point 3 about sampling.4. Explain and correct the error mentioned by Reviewer 1's point 4 (truncation of the horizontal axis).5. Reviewer 2 has raised serious concerns with your handling of race and racial stereotypes. Your response to their critique should be very thorough and detailed.6. Address Reviewer 2's point about obsolete data.7. Both reviewers raise concerns about the ecological validity of your conclusions. That should be remedied (i.e., explained as a limitation) and you should provide a thorough explanation (to the reviewers) of why your data/conclusions are limited in this way yet still important.8. Thoroughly address Reviewer 2's point about ethical concerns (in their point 7). These are just the main issues you need to thoroughly address when submitting your revision. There are other important problems that should also be changed (read both reviewers' comments in detail). Please, do not submit your revised paper without making very substantial changes to it and without providing thorough responses to all the issues raised by the reviewers (not limited to the 8 underscored above). When preparing your revision, keep in mind that one reviewer recommended rejecting your paper (the other one indicated major revision), which means that the reworked paper will take a lot of effort and might very well not be accepted.

We look forward to receiving your revised manuscript.

Kind regards,

Tibor Rutar

Academic Editor

PLOS ONE

Journal Requirements:

Reviewers' comments:

Reviewer's Responses to Questions

**Comments to the Author**

1. Is the manuscript technically sound, and do the data support the conclusions?

Reviewer #1: Partly

Reviewer #2: No

2. Has the statistical analysis been performed appropriately and rigorously? 

Reviewer #1: I Don't Know

Reviewer #2: Yes

3. Have the authors made all data underlying the findings in their manuscript fully available?

Reviewer #1: Yes

Reviewer #2: Yes

4. Is the manuscript presented in an intelligible fashion and written in standard English?

Reviewer #1: Yes

Reviewer #2: Yes

5. Review Comments to the Author

Reviewer #1: The paper examines how attributions of intentionality depend on distal/proximal intent and on group membership. I find the overall topic interesting and worthy of investigation.

The paper is well written, clear and to the point. However, I have some major concerns, mainly about the attribution of the results to actors’ relative group affiliation (ingroup or outgroup), rather than to their absolute group affiliation (white/black, liberal/conservative).

In sum, I suggest to collect data that will allow to assess in/out group effects, or to reframe the paper to deal with specific groups (rather than with outgroup members in general).

Major comments:

1. In the main text, the scenario in Study 1 is about Alex trying to kill Linda by drowning. In the appendix the scenario is about JG trying to kill his uncle by running him over with his car.

2. The actors in Study 1 are either Black or White men. The Mturk sample hardly includes Black participants. Would the results differ for Black participants assessing the intentionality of Black/White actors? If not, it would suggest that the results are not about the actor belonging to an outgroup, but about the actor being black. This is a crucial point, because the paper is about “outgroup actors”.

3. The ABPC scale goes from 1 to 7. In Figure 5, which presents the distribution of ABPC scores, the horizontal axis terminates at 5, giving the *false* impression that the “conservative” tertile is somewhat above the center of the scale (i.e., is somewhat conservative). But in reality, there are hardly any participants with an ABPC higher than 4, suggesting that the “conservative” tertile is actually more liberal than conservative.

This leads to a similar situation as in Study 1. In the absence of conservative participants, results cannot be attributed to the relative group affiliation of the actors (i.e., whether they belong to the ingroup or the outgroup), but may be due to actors’ absolute group affiliation (i.e., whether they are liberal or conservative).

Minor comments:

1. Add page and line numbers

2. “In other words, motivated cognition is often most evident in instances when ambiguity provides plausible deniability, or the illusion of objectivity (e.g., 23)” – not clear, sentence seems out of place.

3. Add error bars to figures

4. The figures would be easier to understand if arranged in a 2x2 table, with row labeled “High Distal Intent”/”Low Distal Intent”, and columns labeled “High Proximate Intent”/”Low Proximate Intent”

5. “Second, political intergroup bias may be stronger than racial intergroup bias” – this seems too general. It is based on the current evidence about *moral* reasoning and attribution. Since political groups are plausible construed as based on moral differences, it may be that other forms of bias—not/less related to moral issues—will not display a similar pattern.

6. Homicide is a very specific and extreme moral violation, which is likely not relevant to most people’s daily lives. I find it hard to draw general conclusions from this scenario.

7. The scenario starts with “JG told his friends that he wanted to kill his rich uncle…”. Why not “JG wanted to kill his rich uncle…”? What is the added value of JG telling his friends about his plan?

Reviewer #2: The manuscript "Observers’ Motivated Sensitivity to Outgroup Actors’ Intent" delves into the intricate dynamics of intentionality and moral judgment, examining how the attribution of intention varies based on the actor's group affiliation, either racial or political. This review integrates the provided points, scrutinising the paper’s conceptual depth, handling of sensitive topics such as race and political affiliation, and its theoretical contributions.

1) Conceptual Superficiality and Gradation of Intentionality

The manuscript's exploration of intentionality, particularly the proximal and distal intent framework (PIDI), attempts to offer a nuanced understanding of how intentions are perceived and judged. However, this exploration falls short of a truly gradational analysis, failing to convincingly argue for the distinctiveness and utility of the PIDI framework beyond existing theories in moral philosophy. The delineation between proximal and distal intent, while intended to enrich the understanding of intentionality, is not clearly or convincingly distinguished from pre-existing theories, which already address the complexity of intentions and their moral evaluations. Such judgements of intent also form the basis of most legal systems and sentencing, yet no reference is made to the vast body of research in criminology or criminal psychology. While this bidimensional approach is a commendable attempt to refine our understanding of how intention is perceived, the manipulations of proximal intent often come across as convoluted, and while the model promises a greater granularity and gradation in exploring the perceptions of intent, it only offers four categorical options, hence missing out on the promise of a more nuanced, dimensional model or exploring the possible variations of intentionality within the set categories. There is also no mention of the possible plurality of "distal" intents, which may converge or conflict in a situation ("proximal" intents seem entirely anchored in action and thus sequential). Finally, the claim that "proximal" and "distal" intents are completely independent remains questionable.

2) Insensitive Handling of Race and Stereotypes

The study's approach to racial dynamics, particularly through the framing of racial group membership in Study 1, raises concerns. Utilising race as a variable in scenarios of moral transgression without a nuanced or critical understanding of racial dynamics can inadvertently reinforce stereotypes. By relying on racial stereotypes and dated instruments (such as the use of the Attitudes Toward Blacks Scale (ATBS), which was initially developed in the 1970s as “Attitude Toward Negroes Scale”) and the assumptions that South Asian participants will identify with "the White ingroup" and judge Black people as "outgroup members", the study inadvertently reinforces simplistic and monolithic views of racial groups, failing to acknowledge the complex, intersectional identities and experiences of individuals within these groups. This oversight is additionally evident in the failure to include Black people as participants in Study 1, hence always positioning White (or non-Black, which is often conflated with White, and sometimes used to include Asian participants) participants as the judges of the behaviour of Black (or White) actors. Furthermore, this reinforces the default White gaze, and does not explore the possible biases in moral judgements of White and Black actors from the perspective of Black participants. It is also interesting that in a scenario designed to test the differential perceptions of intent on the basis of racial identity, the racial identity of the character Linda is never disclosed. That of course does not mean that participants imagined her as raceless, and what identity they attributed to her might have affected the severity of their judgements in various ways.

Finally, the data is 10 years old, and the last decade has been pivotal for the changing landscape of racial relations in North America (the last term of Obama's presidency, Trump's presidency, Charlottesville Rally, Ferguson, the Black Lives Matter movement, etc.). The paper never takes the effort to contextualise racial attitudes in the broader sociocultural and historical context, treating them as purely individual variables. They also appear to take a rather simplistic or reductionist perspective on racism, not taking into account the intersectionality and systemic factors. For example, on page 14, the authors assert that "A voluminous psychological literature has demonstrated that people display a range of motivated and non-motivated biases when processing information about racial outgroups, especially young, Black men (e.g., 24-26)" – it is unclear whether there is the greatest body of research on the biases towards young, Black men, or whether the research shows the greatest extent of biases targeting young, Black men; there is no information about whether this relates to a criminal setting (as in the study) or other settings (research suggests greater biases against Black women in workplace and healthcare settings, in judgements of attractiveness, in social and economic marginalisation, etc.; not to mention other intersecting axes of marginalisation, such as disability, class, education,...). This is just an example of the way that the authors use quick and convenient statements to justify their methodological choices (showing a picture of a young Black/White actor to explore the differences in perceptions of intent) without acknowledging context, intersectionality, or cultural and narrative templates circulating around racial criminal stereotypes (especially in North America). And while contemporary (liberal) racism tends to operate along quite subtle lines, the authors regurgitate 18th century level discourse about the mental capacities of Black people (e.g., “Study 1 data suggest that non-Black participants assume that Black actors possess the capacity to hold a broader aim (distal intent) in mind.” (p. 20)). While the authors pose the hypothesis on the basis that "Black individuals are more strongly governed by immediate, impulsive concerns, rather than abstract, rational aims (27-29)" (p. 14), attributing such deeply racist beliefs to their hypothetical participants, the repetition of such claims in scientific literature can serve to validate and reinforce these highly problematic views.

3) Theoretical Contributions and Distinctions from Existing Theories

The paper struggles to articulate the unique contributions of the PIDI framework and its distinction from existing moral psychological and philosophical theories. The notion that global or overarching motives (distal intent) are considered secondary to specific actions (proximal intent) in moral judgments is not adequately justified or explored, leading to confusion about the framework's theoretical underpinnings and its practical implications for understanding moral responsibility. Many assumptions are made about the psychological processes underpinning moral judgements, or the phenomenology of moral reasoning, yet no substantial evidence or rationale is provided. In the manuscript, there is the constantly recurring fallacy of inferring processes from outcomes without sufficient empirical or theoretical grounding.

4) Sensitivity to Context and Narrative Construction

The construction of scenarios for judging moral transgressions lacks justification and sensitivity to broader social and cultural contexts. By relying on questionable narrative templates and stereotypes (such as men resorting to extreme physical violence against women, women scheming against husbands and committing premeditated murder), the study overlooks the rich tapestry of moral reasoning that individuals employ, which is shaped by diverse social, cultural, and personal experiences. The studies further fail to critically examine or justify the chosen scenarios for moral judgements, which seem more rooted in narratives from TV crime series and novels than in the complex realities of moral decision-making. Additionally, while the racial or political identities are experimentally manipulated, the importance of gender and the accompanying biases and stereotypes is glaringly omitted.

5) Biases and Assumptions

The study's hypotheses and interpretations are heavily influenced by selective previous research and assumptions that may not allow for alternative interpretations of the data. Especially the framing of hypotheses in Study 1 is very problematic; they rely on cherry-picked previous research, and act as templates for interpretation, whereas numerous other hypotheses could be made, and the data could be explained with other, competing theories. Moreover, because the proximal–distal distinction in intentionality is in many cases questionable, and because the quantitative data does not illuminate the underlying psychological mechanisms, the interpretation and discussion often seem dubious. The biases and assumptions are also evident in the framing of moral judgments from the perspective of predominantly white or non-black participants, without adequately considering the perspectives of other racial groups or exploring the potential for motivated racial moral reasoning beyond simplistic binaries.

6) Political Orientation Conceptualisation:

The study’s conceptualisation of political orientation as a binary — distinguishing only between liberals and conservatives — oversimplifies the complexity of political identities. This binary is notably North American-centric and fails to account for the increasing diversity within the political spectrum, even within North America itself. Such an approach neglects rich literature from political psychology and political sciences that could provide nuanced insights into biases and stereotypes about political outgroup members. The lack of acknowledgment of this literature leaves the study with a significant conceptual gap, undermining the applicability of its findings to broader discussions on political ideology and moral judgment.

7) Methodological Concerns, Ecological Validity, and Future Directions

The lack of information on the briefing and debriefing procedures, especially given the studies' sensitive nature, raises ethical concerns. There is also a lack of commentary on how the application of scales assessing the participants' racial attitudes and political orientation might have acted as priming and biased the subsequent results. The claim that the study provides “a standardized, theoretically – and empirically – grounded template for future researchers who wish to measure intergroup judgments involving ambiguously-intentional acts” (p. 32) appears overstated given the limitations of the study's ecological validity and sensitivity to context. The assertion of offering a standardised template for future research is premature, especially considering the unexamined assumptions in the process of "standardisation" and the lack of in-depth study and understanding of the processes of moral reasoning that the PIDI framework purports to illuminate. There is also a lack of discussion on potential applications of the findings, such as understanding biases and judgments in criminal justice contexts, where the perceptions and biases of jury members can have severe effects on people's lives.

Finally, the paper's theoretical and methodological foundations would benefit from greater diversity among the authors and participants, allowing for a broader and more nuanced exploration of the topics at hand.

In conclusion, while the manuscript attempts to contribute to the literature on intentionality and moral judgment, it is hindered by conceptual superficiality, insensitive handling of racial and other group identities, and a lack of clarity in its theoretical contributions. Future research should strive for greater conceptual depth, sensitivity to the complexities of race and identity, and a clearer articulation of how frameworks such as PIDI can enhance our understanding of moral judgments and intentions.

6. PLOS authors have the option to publish the peer review history of their article (what does this mean?). If published, this will include your full peer review and any attached files.

Reviewer #1: No

Reviewer #2: No

---

## [Author Response · Author response to Decision Letter 0]

6 May 2024

Response to the Editor:

1. Collect more data or reframe the paper in line with Reviewer 1's recommendation.

With a good deal of new text throughout the manuscript, we have significantly reframed the paper to address Reviewer 1’s question about whether the effects are generic ingroup/outgroup effects or effects that are specific to the groups in question. We have been persuaded to reframe the effect as one about stereotypes about specific stigmatized groups - African Americans, Conservatives, and Liberals - rather than a general ingroup/outgroup effect. We have also changed the title of the paper so that it no longer refers to ‘outgroups’, but instead to specific stigmatized groups. We now offer more measured, modest claims about what our data do and do not imply.

2. Correct the error pointed out by Reviewer 1 in their point 2 (having to do with the mismatch between the presented scenario and the appendix description).

We have now done so.

3. Take heed of Reviewer 1's point 3 about sampling.

We have done so. I provide more specifics below in the section that responds to Reviewer 1.

4. Explain and correct the error mentioned by Reviewer 1's point 4 (truncation of the horizontal axis).

We have now done so.

5. Reviewer 2 has raised serious concerns with your handling of race and racial stereotypes. Your response to their critique should be very thorough and detailed.

We have added a significant amount of new text throughout the manuscript that addresses these concerns. We provide more detail below in our response to Reviewer 2. But to briefly preview, we suggest that most of Reviewer 2’s concerns about how we operationalized race, though legitimate and worthy, require responses that are well beyond the scope of this paper. That is because this is not a paper about race and racism. It is a paper about judgments of intent. We simply used the actor’s race - in one study - as a device to manipulate participants’ interpretation of the actor’s action. This manipulation theoretically could have been done in any number of ways (for example, by varying participants’ financial incentives). The nuanced and complex issues about intersectionality and sociocultural influences that the reviewer raises are important areas of study but are tangential to our primary focus.

6. Address Reviewer 2's point about obsolete data.

As we note below, this criticism is more relevant to a paper that purports to be a detailed examination of the psychology of racism in 2024. Our paper does not have that goal. Instead, as noted above, our race manipulation in one study was a means to an end, not an end unto itself. 

7. Both reviewers raise concerns about the ecological validity of your conclusions. That should be remedied (i.e., explained as a limitation) and you should provide a thorough explanation (to the reviewers) of why your data/conclusions are limited in this way yet still important.

We have added a lengthy new section to the General Discussion (pp. 28-29) that more clearly identifies ecological validity concerns and explains that we constructed our scenarios with the full understanding of their limited ecological validity. We go on to argue that this approach is often used in both philosophy and psychological research because such scenarios may help to identify and isolate specific, fundamental building blocks of moral cognition.

8. Thoroughly address Reviewer 2's point about ethical concerns (in their point 7).

 The reviewer writes, “Utilising race as a variable in scenarios of moral transgression without a nuanced or critical understanding of racial dynamics can inadvertently reinforce stereotypes.”

We respond to this point in detail below, but to foreshadow, we remind the reviewer that our studies received approval from our university’s rigorous Research Ethics Board (REB) - as have presumably all published papers that in the recent history of social psychology that have used similar manipulations of a target person’s race. We would humbly suggest that it is the job of such ethics review boards (composed of experts in scientific ethics) to make such ethical judgments. We received REB approval for these studies and have submitted the requisite evidence to PLOS One when we submitted the manuscript. In other words, an impartial body has determined that the risk of inadvertently reinforcing stereotypes is negligible and is outweighed by the potential scientific benefit.

Response to Reviewer #1: 

Major comments:

1. In the main text, the scenario in Study 1 is about Alex trying to kill Linda by drowning. In the appendix the scenario is about JG trying to kill his uncle by running him over with his car.

Thank you for pointing out this oversight. It has now been corrected.

2. The actors in Study 1 are either Black or White men. The Mturk sample hardly includes Black participants. Would the results differ for Black participants assessing the intentionality of Black/White actors? If not, it would suggest that the results are not about the actor belonging to an outgroup, but about the actor being black. This is a crucial point, because the paper is about “outgroup actors”.

We have changed the text throughout (pp. 7,8,14,17,18,24,26,27) so that it refers to conclusions regarding participants’ judgments of the Black actor, the conservative actor, the liberal actor, or ‘members of stigmatized groups’ (rather than using the language of ‘ingroup/outgroup’).

3. The ABPC scale goes from 1 to 7. In Figure 5, which presents the distribution of ABPC scores, the horizontal axis terminates at 5, giving the *false* impression that the “conservative” tertile is somewhat above the center of the scale (i.e., is somewhat conservative). But in reality, there are hardly any participants with an ABPC higher than 4, suggesting that the “conservative” tertile is actually more liberal than conservative.

Our implementation of the ABPC used a scale with a range of 1-5. The ABPC in the appendix is the original ABPC, differing only in its use of a scale with a range of 1-7. Nonetheless, we have now changed our language to read ‘relatively conservative’ and ‘relatively liberal’. We now acknowledge even more explicitly in the text that the distribution of political orientation scores did not use the entire scale. Nonetheless there was enough variability in the data to allow clear patterns to emerge. As we now state on p. 23-24, this means that the effects we report may, in fact, underestimate the true magnitude of the difference between liberal and conservative participants.

Minor comments:

1. Add page and line numbers

We have now done so.

2. “In other words, motivated cognition is often most evident in instances when ambiguity provides plausible deniability, or the illusion of objectivity (e.g., 23)” – not clear, sentence seems out of place.

We have added additional text and citations on pp. 14-15 explaining that this is a canonical principle of motivated cognition. Decades of studies indicate that people generally only engage in motivated cognition when there is little danger of being exposed as biased or delusional.

3. Add error bars to figures

We have now done so.

4. The figures would be easier to understand if arranged in a 2x2 table, with row labeled “High Distal Intent”/”Low Distal Intent”, and columns labeled “High Proximate Intent”/”Low Proximate Intent”

We have now arranged the bar graphs into the 2x2 table format you suggest.

5. “Second, political intergroup bias may be stronger than racial intergroup bias” – this seems too general. It is based on the current evidence about *moral* reasoning and attribution. Since political groups are plausible construed as based on moral differences, it may be that other forms of bias—not/less related to moral issues—will not display a similar pattern.

We are persuaded by the reviewer. We have changed the sentence to read “political intergroup may be considered more morally justified than racial intergroup bias”.

6. Homicide is a very specific and extreme moral violation, which is likely not relevant to most people’s daily lives. I find it hard to draw general conclusions from this scenario.

We have added additional text on p. 29 noting that the results of these studies may vary if the actor committed, for example, a white-collar crime.

Response to Reviewer #2: 

1) Conceptual Superficiality and Gradation of Intentionality

The manuscript's exploration of intentionality, particularly the proximal and distal intent framework (PIDI), attempts to offer a nuanced understanding of how intentions are perceived and judged. However, this exploration falls short of a truly gradational analysis, failing to convincingly argue for the distinctiveness and utility of the PIDI framework beyond existing theories in moral philosophy. The delineation between proximal and distal intent, while intended to enrich the understanding of intentionality, is not clearly or convincingly distinguished from pre-existing theories, which already address the complexity of intentions and their moral evaluations.

We very much appreciate the reviewer’s exhortation to argue convincingly that the PIDI is conceptually distinct from previous models. Unfortunately, the reviewer does not name a single example of a pre-existing psychological theory that might contain excessive overlap with ours. Nonetheless, this comment spurred us to examine the psychology literature further. On pp. 5-6, we have enriched the literature review with several additional pre-existing accounts of laypeople’s theories of intentionality. We spell out more explicitly how our approach differs from (and builds upon) others, especially the influential Malle & Knobe (1997) model. For instance, on p. 6, we now write:

We view the PIDI framework as extending several of Malle and Knobe’s elements by introducing proximal versus distal variants. For example, Malle and Knobe

operationalized intention as “intending, meaning, deciding, choosing, or planning to

perform the act” (p. 107). The PIDI model asks, what is the actor’s subjective

representation of the ‘act’? For example, does he think he is ‘pulling a trigger’ or

‘murdering his uncle’? Similarly, consider the concept of “desire.” Malle and Knobe

(1997) defined desire as “a wish for a particular outcome” (p. 106). The PIDI model asks,

what exact desire is forefront in the actor’s mind, pulling the trigger or killing the uncle?

By now, numerous studies have provided evidence that perceivers are quite sensitive to

these distinctions (15,16,18). 

Such judgements of intent also form the basis of most legal systems and sentencing, yet no reference is made to the vast body of research in criminology or criminal psychology.

We had initially excluded criminology and legal theory citations in order to keep the paper more focused on the psychological literature, but we have now re-inserted a range of relevant legal and philosophical works (e.g., pp. 27-28) that have, in fact, significantly informed our thinking. As now noted in the text, we have been particularly influenced by the writings of legal theorists RA Duff and HLA Hart. These philosophical theories of mens rea, however, have rarely been put to the empirical test. That is what our research program sets out to address (not only in this paper, but several others; see Plaks & Robinson, 2017 for an extensive review).

While this bidimensional approach is a commendable attempt to refine our understanding of how intention is perceived, the manipulations of proximal intent often come across as convoluted, and while the model promises a greater granularity and gradation in exploring the perceptions of intent, it only offers four categorical options, hence missing out on the promise of a more nuanced, dimensional model or exploring the possible variations of intentionality within the set categories.

We now state even more explicitly that we view PI and DI as continuous dimensions. For example, on pp. 4-5, we now state:

It is important to note that we consider the presence or absence of proximal intent and

distal intent as continua, not dichotomies. Thus, at a given moment, an actor’s distal

intent may be more prominently in mind than her proximal intent, but that does not imply

that the proximal intent is absent.

In the present studies, we operationalize proximal and distal intent in a dichotomous fashion to permit the manipulation of both in a standard, social psychological 2X2 full factorial design. Indeed, whereas other psychological theories of intent often dichotomize intent (e.g., Schein & Gray, 2018), we explicitly do NOT view these dimensions as binary constructs. We are very clear about this in several places, including pp. 4, 29. Consistent with over 80 years of social psychological research, we use a 2X2 design with the understanding that doing so is a model of reality - not reality itself.

There is also no mention of the possible plurality of "distal" intents, which may converge or conflict in a situation ("proximal" intents seem entirely anchored in action and thus sequential).

Thanks to the reviewer’s astute comment, we now highlight that, for example, when J.G. presses the accelerator in the Proximal Intent Higher condition, this is a case when a second, different distal intent (avoid hitting pedestrian) has precedence in the actor’s mind - rather than the morally/legally-relevant distal intent (kill his uncle) (p. 5). On p. 28-29, we have added new text describing how the famous legal case of Oscar Pistorius represents an example of an actor who claimed to be acting with full proximal intent (pull trigger with awareness and control), but with a distal intent (kill intruder) that was different from the distal intent with which he was charged (kill girlfriend). We thank the reviewer for encouraging us to clarify that when we say, “Distal Intent Low” that is actually shorthand for “focal Distal Intent Low”.

Finally, the claim that "proximal" and "distal" intents are completely independent remains questionable. 

We agree. We never state that they are completely independent and do not wish the reader to infer that they are completely independent. We have added language such as “somewhat independent” and “partially independent” throughout the paper (e.g., pp. 4,7,27). It is worth noting that in these studies (as well as virtually every other study we have ever run using the PIDI framework (N > 15,000), the data repeatedly reveal two main effects (DI, PI), but no DIxPI interaction. This indicates that, empirically speaking, PI and DI do enjoy a significant degree of independence.

2) Insensitive Handling of Race and Stereotypes

The study's approach to racial dynamics, particularly through the framing of racial group membership in Study 1, raises concerns. Utilising race as a variable in scenarios of moral transgression without a nuanced or critical understanding of racial dynamics can inadvertently reinforce stereotypes.

We would respectfully suggest that, for decades, experimental social psychologists have purposely created simplified, operational models of reality to manipulate their variables of interest in specific studies. But the creators of such studies typically maintain the full understanding that these are simplified models. Importantly, we would also note that countless institutional ethics review committees (including our own) have determined that the likelihood that participants will become more racist by participating in a study that varies the actor’s race is negligible and outweighed by the potential scientific insights. As per PLOS ONE policies, we have furnished our institutional ethics review approval. This board of experts in ethics and law has scrutinized our proposal, considered the possible harm to participants, and decided that any potential harm falls within the latitude of everyday experience. To learn more about the ethics review process at our university, please see: https://research.utoronto.ca/ethics-human-research/research-ethics-boards.

 By relying on racial stereotypes and dated instruments (such as the use of the Attitudes Toward Blacks Scale (ATBS), which was initially developed in the 1970s as “Attitude Toward Negroes Scale”)

We would respectfully suggest that we did not ‘rely’ on the ATBS scale. In fact, 

---

## [Decision Letter · Decision Letter 1]

11 Jun 2024

Observers’ Motivated Sensitivity to Stigmatized Actors’ Intent

PONE-D-23-26287R1

Dear Mr. Staples (M.A.),

We’re pleased to inform you that your manuscript has been judged scientifically suitable for publication and will be formally accepted for publication once it meets all outstanding technical requirements.

Kind regards,

Tibor Rutar

Academic Editor

PLOS ONE

Additional Editor Comments (optional):

I'd like to apologize for the long delays you've had to endure during the review process. I received your manuscript in late November. I then immediately started inviting reviewers in the field of social psychology of inter-/intra-group dynamics and intentionality. Between November and January, roughly 40 reviewers were invited. Only 2 accepted my invitation, the others either declined or were non-responsive. Things then got more complicated due to two reasons. First, one of the two reviewers became, despite constant reminders, evasive and in the end very late with their review. This ended up severely delaying the whole process. Second, during the second stage of reviews in May, the other reviewer initially accepted my invitation to review your revised version of the manuscript, but failed to turn in their review. I tried reinviting them twice, but the reviewer became non-responsive. To solve the problem, I invited a new, third reviewer, who carefully read your initial submission, both reviewers' comments, your response to the comments, and your revised manuscript. 

Reviewers' comments:

Reviewer's Responses to Questions

**Comments to the Author**

1. If the authors have adequately addressed your comments raised in a previous round of review and you feel that this manuscript is now acceptable for publication, you may indicate that here to bypass the “Comments to the Author” section, enter your conflict of interest statement in the “Confidential to Editor” section, and submit your "Accept" recommendation.

Reviewer #3: All comments have been addressed

2. Is the manuscript technically sound, and do the data support the conclusions?

Reviewer #3: Yes

3. Has the statistical analysis been performed appropriately and rigorously? 

Reviewer #3: Yes

4. Have the authors made all data underlying the findings in their manuscript fully available?

Reviewer #3: Yes

5. Is the manuscript presented in an intelligible fashion and written in standard English?

Reviewer #3: Yes

6. Review Comments to the Author

Reviewer #3: I believe the authors have addressed all previous suggestions adequately. As noted in the review, the suggestions that were not directly incorporated in the revised paper are extensively commented on by the authors who satisfactorily justify their decision not to address the concerns raised by previous reviwers. The paper meets scientific standards as applied in other experimental studies in psychology and represents a valuable contribution to the journal.

7. PLOS authors have the option to publish the peer review history of their article (what does this mean?). If published, this will include your full peer review and any attached files.

Reviewer #3: **Yes: **Minea Rutar
